**Data Availability Statement:** All relevant data are within the paper and its Supporting Information files.

# Antifungal therapy in the management of fungal secondary infections in COVID-19 patients: A systematic review and meta-analysis

Sujit Kumar Sah[1], Atiqulla Shariff[1], Niharika Pathakamuri[2], Subramanian Ramaswamy [3]*, Madhan Ramesh[1], Krishna Undela[4], Malavalli Siddalingegowda Srikanth[1], Teggina Math Pramod Kumar[5]

1 Department of Pharmacy Practice, JSS College of Pharmacy, JSS Academy of Higher Education and Research, Mysuru, Karnataka, India, 2 Department of Pharmacy Practice, TVM College of Pharmacy, Rajiv Gandhi University of Health Sciences, Ballari, Karnataka, India, 3 Department of Rheumatology & Immunology, JSS Hospital, JSS Academy of Higher Education and Research, Mysuru, Karnataka, India, 4 Department of Pharmacy Practice, National Institute of Pharmaceutical Education and Research (NIPER), Guwahati, Assam, India, 5 Department of Pharmaceutics, JSS College of Pharmacy, JSS Academy of Higher Education and Research, Mysuru, Karnataka, India

* subsan05@gmail.com

## Abstract

### Objectives

The prevalence of fungal secondary infections among COVID-19 patients and efficacy of antifungal therapy used in such patients is still unknown. Hence, we conducted this study to find the prevalence of fungal secondary infections among COVID-19 patients and patient outcomes in terms of recovery or all-cause mortality following antifungal therapy (AFT) in such patients.

### Methods

We performed a comprehensive literature search in PubMed®, Scopus®, Web of Sciences™, The Cochrane Library, ClinicalTrial.gov, MedRxiv.org, bioRxiv.org, and Google scholar to identify the literature that used antifungal therapy for the management fungal secondary infections in COVID-19 patients. We included case reports, case series, prospective & retrospective studies, and clinical trials. Mantel Haenszel random-effect model was used for estimating pooled risk ratio for required outcomes.

### Results

A total of 33 case reports, 3 case series, and 21 cohort studies were selected for final data extraction and analysis. The prevalence of fungal secondary infections among COVID-19 patients was 28.2%. Azoles were the most commonly (65.1%) prescribed AFT. Study shows that high survival frequency among patients using AFT, received combination AFT and AFT used for >28 days. The meta-analysis showed, no significant difference in all-

**Funding:** The authors received no specific funding for this work.

**Competing interests:** The authors have declared that no competing interests exist.

cause mortality between patients who received AFT and without AFT (p = 0.17), between types of AFT (p = 0.85) and the duration of AFT (p = 0.67).

## Conclusion

The prevalence of fungal secondary infections among COVID-19 patients was 28.2%. The survival frequency was high among patients who used AFT for fungal secondary infections, received combination AFT and AFT used for >28 days. However, meta-analysis results found that all-cause mortality in COVID-19 patients with fungal secondary infections is not significantly associated with type and duration of AFT, mostly due to presence of confounding factors such as small number of events, delay in diagnosis of fungal secondary infections, presence of other co-infections and multiple comorbidities.

## 1. Introduction

Severely ill coronavirus disease-19 (COVID-19) patients, admitted to intensive care units (ICUs) are at increased risk of bacterial and fungal secondary infections. Pulmonary aspergillosis, invasive candidiasis, and mucormycosis are the most frequently reported fungal secondary infections, leading to increased morbidity and mortality in COVID-19 patients [1]. The most common pathogens reported belongs to *Aspergillus*, *Rhizopus and Candida* species. Looking back to 2003, the incidence of fungal secondary infections in COVID-19 patients was high and ranges from 14.8–33% in mild to severely ill patients [2]. However, the recent clinical scenarios from the globe have raised concerns about fungal secondary infections and their management in COVID-19 patients [3]. The presence of diabetes mellitus, cancers, additional immunocompromised status, use of steroids and/or immunosuppressive agents, use of mechanical ventilators are some of the identified risk factors for fungal secondary infections in hospitalized COVID-19 patients [4]. The time interval between COVID-19 diagnosis and the development of fungal co-infection varies widely. In addition, the abrupt development of clinical features makes it more fatal [5]. Therefore, early detection and management help to prevent severe illness and associated deaths.

Currently, there are no established guidelines for the management of fungal secondary infections in COVID-19 patients. However, there are various case reports, case series, and cohort studies published with regard to the management of fungal secondary infections in COVID-19 patients. The commonly used antifungal therapy (AFT) includes liposomal amphotericin B, azole, and echinocandins [6]. Hence, we conducted this study to systematically published literatures to explore the prevalence of fungal secondary infections among COVID-19 patients and outcomes in terms of recovery or all-cause mortality associated with the use of AFT in such patients.

## 2. Methods

Protocol for this study was designed based on the preferred reporting items for systematic review and meta-analysis protocols PRISMA-P 2015 statements [7] and has been registered at PROSPERO (CRD42021259957). A comprehensive study was conducted following the PRISMA 2009 statement for reporting systematic reviews and meta-analysis [8]. We reviewed all the human studies published in English that included patients with a confirmed diagnosis of COVID-19 with fungal secondary infections across all the age groups in whom at least one

antifungal agent was used. The fungal secondary infections were defined as those caused by any fungal species either at admission or during the hospital stay. The fungal species were detected by observing the colony morphology and color of the isolated culture media. The details of inclusion criteria are presented in PICOS format in *S1 Appendix*. We excluded review articles, systematic reviews, meta-analysis, brief reports, short reports, editorials, commentaries, notes, book chapters, abstracts, surveys, conference proceedings, posters presentations, unpublished materials and guidelines.

## 2.1. Data sources

We performed a comprehensive literature search using predefined search terms in eight online search engines namely, PubMed®, Scopus®, Web of Sciences™, The Cochrane Library (Central), ClinicalTrial.gov, MedRxiv.org, bioRxiv, and Google scholar to identify the literature records published between 1st January 2020 and 30th June 2021. A manual hand search of references was also performed to avoid missing any relevant literature. Further, all the literatures retrieved from the search engines were transferred to the Mendeley reference manager to remove duplicate records. The details of search strategies are presented in *S2 Appendix*.

## 2.2. Study selection

The study titles and abstracts were independently screened by two authors to determine whether the studies met the inclusion criteria. The full-text records of these studies were further reviewed for final inclusion. Additionally, the reference section of all the selected articles were hand-searched by another author, to identify the additional literature records for possible inclusions. If any missing study relevant information, review authors were actively participated in the searching for original resources or contacted study authors through mail to obtain missing information. The discrepancies related to the selection and eligibility were resolved through discussion between the first three authors, and unresolved issues were addressed by the 4th and 5th authors. The final decision was made following consensus between all the authors.

## 2.3. Data extraction

Two authors independently performed the data extraction from all the included records and were documented in a specifically designed data extraction tool (©Microsoft excel-2019). The variables such as the first author of the publication, year of publication, geographical location where the study was performed, type of the study (case reports, case series, prospective studies, retrospective studies and clinical trials), sample size, age (in year) and gender, diagnosis of fungal co-infection, types of fungal species isolated / cultured, name of antifungal drugs, type of therapy (mono or combination), dose, frequency & route of administration, total duration of antifungal therapy (in days), total duration of hospital stay (in days), and patient outcomes (either alive or dead) were recorded.

## 2.4. Data synthesis

The outcome measures were to assess the prevalence of fungal secondary infections (cohort studies), all-cause mortality in patients using AFT and without AFT, all-cause mortality associated with type of AFT (mono or combination AFT), and all-cause mortality associated with the duration of AFT ($\leq$28 days or $>$28 days) among COVID-19 patients with fungal secondary infections.

## 2.5. Statistical analysis

A meta-analysis was performed for all the eligible cohort studies. If three or more studies reporting any or similar fungal secondary infections and use of AFT were identified, and applied Mantel Haenszel random-effect model for estimating pooled risk-ratio using Review Manager (RevMan) 5.4.1 software ([Computer program] Version 5.3. Copenhagen: The Nordic Cochrane Centre, The Cochrane Collaboration, 2014) for required outcomes. $I^2$ statistic was used to evaluate the heterogenicity of studies following Cochrane recommendations [9] and heterogenicity was considered substantial if $I^2$ was >50%.

## 2.6. Risk of bias assessment

We used the methodological quality & synthesis guide for evaluating the risk of bias involved in case reports, and case series. Based on the total score, the methodological quality & synthesis guide categorizes the risk of bias as low (5), medium (3–4) and high (0–2) [10]. Whereas, Newcastle-Ottawa Quality scale was used to assess the quality of the cohort studies [11]. A total score of three or less is indicative of poor quality, 4–6 as moderate quality, and 7–9 as high quality of cohort studies [12].

# 3. Results

## 3.1. Study selection

A total of 403 records were identified in the scientific databases and hand search of references. After removing the 102 duplicate records, the remaining 301 records were screened for title and abstract. Of them, 162 records were excluded as they were irrelevant to the study. Further, 139 full text articles were reviewed and 82 articles were excluded from them (*S1 Table*). Finally, 33 case reports [18, 23–54], 3 case series [55–57] and 21 cohort studies [58–78] were selected for final data extraction and analysis (Fig 1).

## 3.2. Study characteristics

The characteristics of the 57 eligible studies is presented in Table 1 (case reports, n = 33 and case series, n = 3) & Table 2 (cohort-studies, n = 21). Most of the studies published were from the five continents that included Europe [n = 24, 42.1%] (Denmark [23], Italy [27, 34, 35], Ireland [29], France [30, 69, 75], Spain [31, 45, 71], Greece [51], Austria [54], Netherlands [55, 57], Germany [58, 63, 67], UK [62, 64, 78], Switzerland [74, 77]), Asia [n = 18, 31.5%] (Iran [24, 28, 68, 76], Iraq [39], Kuwait [33], Qatar [40, 53], Japan [41, 42], India [18, 43, 46, 48, 59], Indonesia [49], China [60], Pakistan [65]), North America [n = 10, 17.5%] (USA [26, 32, 36, 38, 44, 50, 61, 66, 72], Mexico [70]), South America [n = 4, 7%] (Argentina [37, 52, 56] & Brazil [47]) and Australia [25] [n = 1, 1.7%]. All the studies were conducted within the period of January 2020 to June 2021.

## 3.3. Risk of bias

Case report and case series: Based on the methodological quality & synthesis guide, twenty-four (72.7%) case reports were having low [18, 23–31, 33–35, 37, 38, 40, 41, 43–45, 49, 52–54] and nine (27.2%) case reports were having medium risk of bias [32, 36, 39, 42, 46–48, 50, 51]. Whereas, all the three cases series were having low risk of bias [55–57] (*S3 Table*).

Cohort-studies: According to Newcastle-Ottawa Quality Scale, out of 21 studies, twelve studies (57.1%) had a total score between four and six [58, 61, 63, 64, 66, 67, 69, 72, 74, 76–78] and nine studies (42.8%) had a total score of seven [59, 60, 62, 65, 68, 70, 71, 73, 75], indicating moderate and low risk of bias respectively (*S4 Table*).

## PRISMA 2009 Flow Diagram

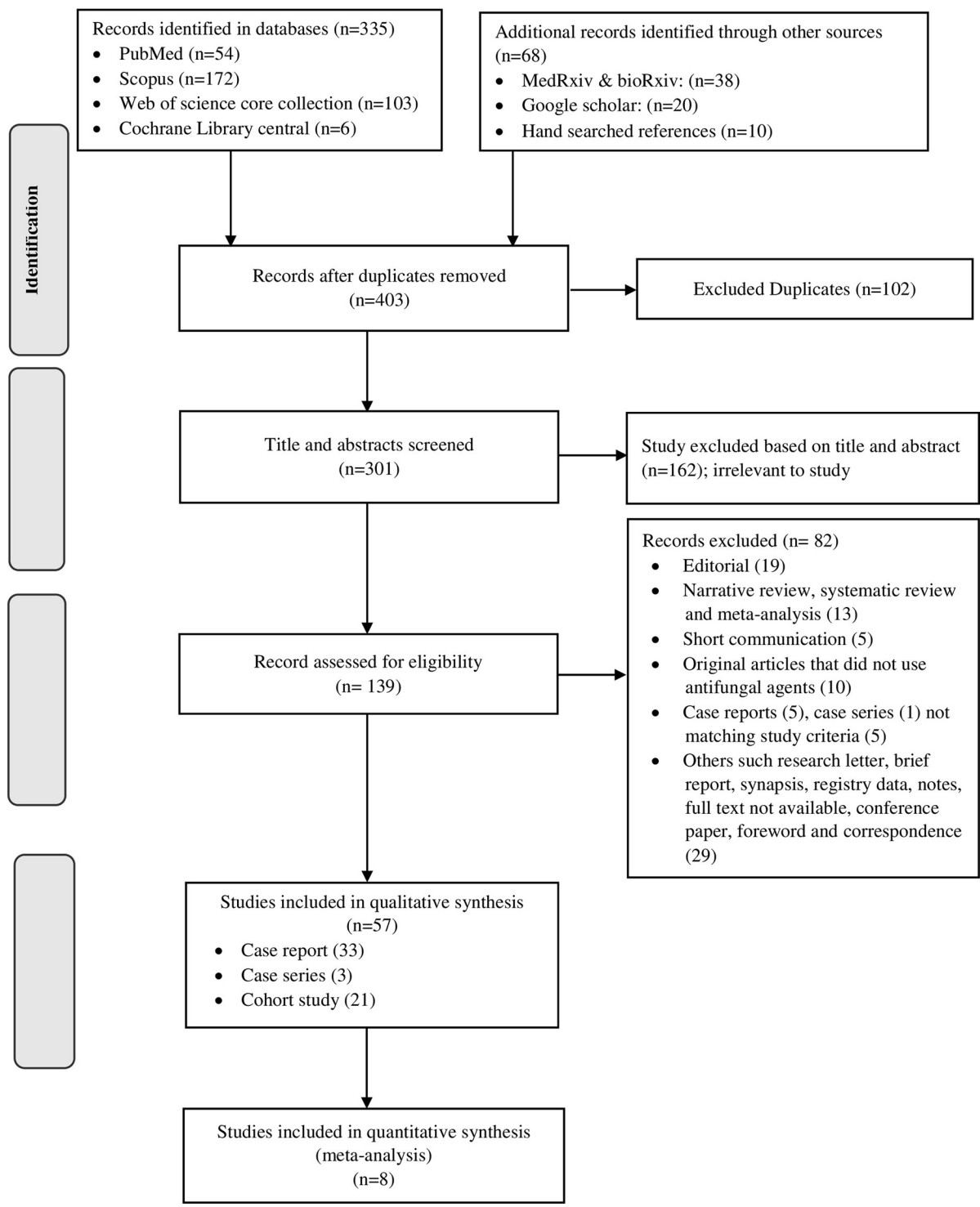

**Fig 1. PRISMA flow diagram of study selection process.**

**Table 1. Study characteristics of case reports and case series studies.**

| First author, Country, Year | Sample size | Age (years) | Gender | Diagnosis | Isolated fungal species in culture (Frequency) | Antifungal therapy (AFT) (Drug, dose, frequency, RoA) | Duration of AFT (days) | #Managed with ICU care or MV support (Yes/No) | Hospital stays (days) | Patient outcome | Overall Risk of bias* |
|---|---|---|---|---|---|---|---|---|---|---|---|
| **Case Report:** | | | | | | | | | | | |
| Haglund A et al, Denmark, 2021 [23] | 1 | 52 | Male | CAPA | A. fumigatus | IV VOR: 300 mg BID then, increased to 400 mg/day, followed by PO VOR 400 mg BID | 90 | Yes | 62 | Alive | Low |
| Hakamifard A et al, Iran, 2020 [24] | 1 | 35 | Male | CAPA | A. ochraceus | IV VOR: (6 mg/kg for first day followed by 4 mg/kg BID) + IV Liposomal AMB: 5 mg/kg/day | 15 | Yes | 15 | Death | Low |
| Sharma A et al, Australia, 2021 [25] | 1 | 66 | Female | CAPA | A. fumigatus | IV VOR: 6 mg/kg loading dose followed by 3mg/kg BID, then PO VOR: 300mg/BID | 18 | Yes | 30 | Recovered | Low |
| Witting C et al, USA,2021 [26] | 1 | 72 | Male | CAPA | A. species | VOR + MIC | 19 | Yes | 80 | Recovered | Low |
| Deana C et al, Italy, 2021 [27] | 1 | 69 | Male | CAPA | A. fumigatus | IV Liposomal AMB: 3 mg/kg | 30 | Yes | 68 | Recovered | Low |
| Nasri E et al, Iran, 2020 [28] | 1 | 42 | Female | CAPA | A. species | IV Liposomal AMB: 5 mg/kg/day | 4 | Yes | 12 | Death | Low |
| Mohamed A et al, Ireland, 2021 [29] | 1 | 66 | Male | CAPA | A. fumigatus + C. albicans | IV Liposomal AMB: 3 mg/kg OD | 7 | Yes | 14 | Death | Low |
| Schein F et al. France, 2020 [30] | 1 | 87 | Female | CAPA | A. species | IV VOR: 6 mg/kg BID at first day, then 4 mg/kg BID | 2 | Yes | 17 | Death | Low |
| Trujillo H et al. Spain, 2020 [31] | 1 | 55 | Female | CAPA | A. fumigatus | PO ISA: 200 mg loading dose of every 8th hourly for 6 doses, followed by 200 mg/day / Nebulized liposomal AMB: 25 mg TID weekly | 20 | Yes | 53 | Recovered | Low |
| Prattes J et al, USA, 2021 [32] | 1 | 70 | Male | CAPA | A. fumigatus | IV VOR: 6 mg/kg BID followed by 4 mg/kg BID | 3 | Yes | 4 | Death | Medium |

(*Continued*)

**Table 1.** (Continued)

| First author, Country, Year | Sample size | Age (years) | Gender | Diagnosis | Isolated fungal species in culture (Frequency) | Antifungal therapy (AFT) (Drug, dose, frequency, RoA) | Duration of AFT (days) | #Managed with ICU care or MV support (Yes/No) | Hospital stays (days) | Patient outcome | Overall Risk of bias* |
|---|---|---|---|---|---|---|---|---|---|---|---|
| Alobaid K et al, Kuwait, 2021 [33] (Reports of 2 cases) | 1 | - | Male | CAPA | A. niger | CAS 70 mg, followed by 50 mg/day; Subsequently VOR 400 mg/ BID followed by 200 mg/day BID | 29 | Yes | 53 | Death | Low |
| | 1 | - | Male | CAPA | A. niger | PO ANI 200 mg then 100 mg, Subsequently Liposomal AMB 350 mg/day | 16 | Yes | 31 | Death | |
| Trovato L et al, Italy,2020 [34] | 1 | 73 | Male | CAPA | A. niger | VOR 800 mg/ day | 2 | Yes | 19 | Death | Low |
| Saccaro LF et al, Italy, 2020 [35] | 1 | 61 | Male | CAPA | A. fumigatus | IV ISA 200 mg BID + IV MIC 100 mg/day BID, followed by IV ISA 200 mg BID | 111 | Yes | 30 | Recovered | Low |
| Bilani N et al, USA, 2020 [36] | 1 | Elderly | Male | Pseudofungi | A. species | VOR 2 dose | NM | Yes | NM | Improved | Medium |
| Fernandez NB et al, Argentina, 2021 [37] | 1 | 85 | Male | CAPA | A. flavus + C. lusitaniae | ANI | 4 | Yes | 44 | Death | Low |
| | | | | | | VOR: 400mg first day, followed by 300 mg/day | NM | | | | |
| Patti RK et al, USA, 2020 [38] | 1 | 73 | Male | CAPA | A. flavus | IV VOR | NM | Yes | 21 | Recovered | Low |
| Kakamad FH et al, Iraq, 2021 [39] | 1 | 50 | Male | CAPA | A. species | Broad spectrum antifungal agents | NM | No | 2 | Recovered | Medium |
| Abdalla S et al, Qatar, 2020 [40] (2 cases) | 1 | 58 | Male | CAPA | A. niger + C. albican | ANI + Liposomal AMB | 1 | Yes | 15 | Death | Low |
| | 1 | 74 | Male | CAPA | A. terreus + C. albican | VOR 400 mg BID | 29 | Yes | 49 | Death | |
| Imoto M et al, Japan, 2021 [41] | 1 | 72 | Male | CAPA | A. fumigatus | MIC 150 mg/ day, next switched to VOR | 9 | Yes | 26 | Death | Low |
| Iwanaga Y et al, Japan,2021 [42] | 1 | 79 | Male | CAPA | A. fumigatus | IV Liposomal AMB | 5 | Yes | 28 | Death | Medium |
| Maini A et al, India, 2021 [43] | 1 | 38 | Male | Sinoorbital CAM | R. oryzae | IV AMB: 300 mg/day; followed by FLU 300 mg | 38 | Yes | 38 | Recovered | Low |

(Continued)

**Table 1.** (Continued)

| First author, Country, Year | Sample size | Age (years) | Gender | Diagnosis | Isolated fungal species in culture (Frequency) | Antifungal therapy (AFT) (Drug, dose, frequency, RoA) | Duration of AFT (days) | #Managed with ICU care or MV support (Yes/No) | Hospital stays (days) | Patient outcome | Overall Risk of bias* |
|---|---|---|---|---|---|---|---|---|---|---|---|
| Khatri A et al, USA, 2021 [44] | 1 | 68 | Male | Cutaneous CAM | R. microsporus | IV Liposomal AMB 550mg/day + PO POS delayed-release 300 mg/day | NM | Yes | 175 | Death | Low |
| Arana C et al, Spain, 2021 (2 cases) [45] | 1 | 62 | Male | Rhinosinusal CAM | R. oryzae | Liposomal AMB + ISA, subsequently POS | 150 | Yes | NM | Recovered | Low |
| | 1 | 48 | Male | Musculoskeletal CAM | L. ramosa | Liposomal AMB 5mg / kg/day + ISA 200 mg TID, then ISA 200 mg TID only | 90 | No | NM | Recovered | |
| Krishna DS et al, India, 2021 (Reports of 2 cases) [46] | 1 | 34 | Male | osteomyelitis and zygoma | Unknown fungal species | IV liposomal AMB 5 mg/kg/day, followed by PO ITR 200 mg | 60 | No | NM | Recovered | Medium |
| | 1 | 50 | Male | CAM of the right maxilla | Mucor species | IV liposomal AMB 250 mg, followed by PO POS 300 mg | 60 | No | NM | Recovered | |
| Garg D et al, India, 2021 [22] | 1 | 55 | Male | Pulmonary CAM | R. microsporus | IV Liposomal AMB: 3 mg/kg/day | 58 | No | 54 | Recovered | Low |
| Junior ESM et al, Brazil, 2020 [47] | 1 | 86 | Male | Gastrointestinal CAM | Mucor species | AFT | NM | Yes | 7 | Death | Medium |
| Revannavar SM et al, India, 2021 [48] | 1 | NM | Female | CAM | R. species | Conventional AMB | 11 | No | 17 | Recovered | Medium |
| Sari AP et al, Indonesia, 2021 [49] | 1 | 54 | Female | CAC | C. tropicalis | IV MIC | 21 | Yes | 25 | Recovered | Low |
| Chang CC et al, USA, 2020 [50] | 1 | 48 | Female | Acute pulmonary Coccidioidomycosis | Culture report negative | Tab. FLU 400 mg daily | NM | No | 5 | Recovered | Medium |
| Ventoulis I et al, Greece, 2020 [51] (Reports of 2 cases) | 1 | 76 | Male | Saccharomyces cerevisiae | S. cerevisiae | ANI, followed by FLU | 24 | Yes | 8 | Recovered | Medium |
| | 1 | 73 | Male | Saccharomyces cerevisiae | S. cerevisiae | ANI, followed by FLU | 21 | | NM | Recovered | |
| Bertolini M et al, Argentina, 2020 [52] | 1 | 43 | Male | Disseminated histoplasmosis | H. capsulatum | IV AMB: 1mg/kg/day, Switched oral ITR 200mg TID, then 200mg BID | 23 | No | 17 | Recovered | Low |
| Khatib MY et al, Qatar, 2020 [53] | 1 | 60 | Male | Cryptococcemia | Cryptococcus neoformans | ANI 200 mg OD | 38 | Yes | 30 | Death | Low |
| | | | | | C. glabrate | IV AMB 300 mg OD + FLUC 500 mg BID | | | | | |

*(Continued)*

**Table 1.** (Continued)

| First author, Country, Year | Sample size | Age (years) | Gender | Diagnosis | Isolated fungal species in culture (Frequency) | Antifungal therapy (AFT) (Drug, dose, frequency, RoA) | Duration of AFT (days) | #Managed with ICU care or MV support (Yes/No) | Hospital stays (days) | Patient outcome | Overall Risk of bias* |
|---|---|---|---|---|---|---|---|---|---|---|---|
| Seitz T et al, Austria, 2020 [54] | 1 | 72 | Male | CAC | C. glabrata. | CAS | 14 | Yes | 40 | Recovered | Low |
| **Case Series:** | | | | | | | | | | | |
| Meijer EFJ et al, Netherlands, 2020 [55] | 13 | 67.30 (mean) | 76.9% (Male) | CAPA (13) | A. fumigatus (10) | VOR (7), CAS + L-AMB (1), CAS + VOR + L-AMB (4), CAS + VOR (1) | NM | Yes (all patients) | 30 (20–41) | Death (6) / Alive (7) | Low |
| Benedetti MF et al, Argentina, 2021 [56] | 5 | 57 (33–69) | 80% (Male) | CAPA (5) | A. fumigatus (3) | VOR (4), AMB (1), FLU (1) <br><br> IV VOR: 400 mg BID on first day, then 200 mg BID; IV AMB: 5 mg/kg/day <br><br> FLU: 200 mg (loading dose) followed by 100 mg/day | NM | Yes (all patients) | 12±11.76 | Death (1) / Alive (4) | Low |
| Flikweert AW et al, Netherlands, 2020 [57] | 7 | 73 (mean) | 71.4% (Male) | CAPA (7) | A. fumigatus (2) | VOR + ANI (6) | NM | Yes (all patients) | 74 (58–83) | Death (3) / Alive (4) | Low |

#Patient who received ICU care or MV support during hospital stay, anytime

AFT: Antifungal therapy; AMB: Amphotericin B; ANI: Anidulafungin; BID: bis in die (twice daily); BSAA: Broad spectrum antifungal agents; CAS: Caspofungin; CAC: COVID-19 Associated Candidemia; CAPA: COVID-19 Associated Pulmonary Aspergillosis; CAM: COVID-19 Associated Mucormycosis; ECH: Echinocandins; FLU: Fluconazole; FLUC: Flucytosine; IBR: Ibrexafungin; ICU: Intensive care unit; ISA: Isavuconazole; ITR: Itraconazole; IV: Intravenous; MIC: Micafungin; MV: Mechanical ventilation; NM: Not mentioned; NR: Not reported; NYS: Nystatin; OD: Once in a day; PO: per oral (Orally); POS: Posaconazole; RoA: Route of Administration; TID: ter in die (Thrice daily); VOR: Voriconazole.

## 3.4. Participants' characteristics and clinical diagnosis

A total of 1537 patients' data [case report (n = 38), case series (n = 25) and cohort-studies (n = 1474)] was analysed from the included studies. Overall, 479 patients were identified with fungal secondary infections.

**3.4.1. Case report and case series.** Among 38 patients in case reports, 21 (55.2%) patients were diagnosed with COVID-19 associated pulmonary aspergillosis (CAPA) [23–35, 37, 42], nine (23.6%) patients with COVID-19 assicated mucormycosis (CAM) [18, 43–48], two (5.2%) patients with COVID-19 associated candidemia (CAC) [49, 54], and six (15.7%) patients with other fungal secondary infections [36, 46, 50–52, 54]. In case series, all 25 patients were diagnosed with CAPA [55–57] (Table 1).

**3.4.2. Cohort studies.** Out of 1474 patients in cohort-studies, 416 were identified with the fungal secondary infections, accordingly the prevalence of fungal co-infection in cohort-studies was 28.2% (416/1474). A majority [280/416, 67.3%] of these patients were diagnosed with CAPA [58, 61, 63–65, 67, 69–75, 77], followed by CAC [58, 60, 66, 68, 69, 74, 76, 77] [112/416, 26.9%] and CAM [59, 69] [6/416, 1.4%] (Table 2).

**Table 2. Study characteristics of cohort studies (observational studies & retrospective studies).**

| First author, Country, Year | Sample size | Age [mean ±sd/ Median (IQR)] years | Gender Male (%) | Diagnosis (Frequency) | Isolated fungal species in culture (Frequency) | Antifungal therapy (AFT) [Drug, dose, Frequency, RoA] | Duration of AFT (days) | #Managed with ICU care or MV support (Yes/No) | Duration of hospital stays [mean±sd/ Median (IQR)] days | Patient outcome (Frequency) | Overall Risk of bias* |
|---|---|---|---|---|---|---|---|---|---|---|---|
| Rothe K et al, Germany, 2021 [58] | 140 | 63.5 | 64.3% | CAPA (9) | A. fumigatus (9) | ECH (5), VOR (4), FLU (6), Liposomal AMB (8) | NM | NM clearly | 19 (1–47) | Death (18) | Moderate |
| | | (17–99) | | CAC (3) | C. albicans (3) | | | | | Alive (122) | |
| | | | | | | | | | | [Discharged: (95) | |
| | | | | | | | | | | Continue: (27)] | |
| Sen M et al, India, 2021 [59] | 6 | 60.5±12 | 100% | CAM (6) | Mucor species (6) | POS +liposomal AMB (4), POS + liposomal AMB + VOR (1), AMB (1) | ≤28 days: (2) | No | NM | Alive (6) | Low |
| | | | | | | | >28 days: (4) | | | | |
| | | | | | | Loading dose: POS 300 mg BID for first day | | | | | |
| | | | | | | Maintaining dose: POS: 300 mg/day, followed by IV Liposomal AMB: 5–10 mg/ kg/day | | | | | |
| Chen N et al, China, 2020 [60] | 99 | 55.5 ±13.1 | 68% | CAC (4) | C. glabrata (1) | AFT (15) | NM | NM clearly | NM | Death (11) | Low |
| | | | | | C. albicans (3) | | | | | Alive (88) | |
| | | | | | | | | | | [Discharge: (31) | |
| | | | | | | | | | | Continue: (57)] | |
| Permpalung N et al, USA, 2021 [61] | 396 | -64.5 | 58.15% | CAPA (39) | A. species (11) | Antifungal therapy (28) | NM | NM clearly | 41.1 | Death (22) | Moderate |
| | | (54–71) | | | | No antifungal therapy (11) | | | (20.5– 72.4) | Alive (17) | |
| White PL et al, UK, 2020 [62] | 135 | 57 | 2.2 | Yeast infection (17) | C. albicans (13) | FLU (6), VOR (1), CAS (2) | NM | Yes | 19.5 | Death (8) | Low |
| | | (48–64) | | | C parapsilosis (1) | | | (all patients) | (12.3– 33.3) | Alive (9) | |
| | | | | | C. albicans + C. parapsilosis (1) | CAS+ liposomal AMB (1) | | | | | |
| | | | | | | CAS + FLU (2) | | | | | |
| | | | | | Rhodotorula (1) | CAS + VOR (1) | | | | | |
| | | | | | Unclassified Yeast (1) | FLU+ VOR (1) | | | | | |
| | | | | | | FLU+ AMB (1) | | | | | |
| | | | | | | No antifungal therapy (2) | | | | | |

(*Continued*)

**Table 2.** (Continued)

| First author, Country, Year | Sample size | Age [mean ±sd/ Median (IQR)] years | Gender Male (%) | Diagnosis (Frequency) | Isolated fungal species in culture (Frequency) | Antifungal therapy (AFT) [Drug, dose, Frequency, RoA] | Duration of AFT (days) | #Managed with ICU care or MV support (Yes/No) | Duration of hospital stays [mean±sd/ Median (IQR)] days | Patient outcome (Frequency) | Overall Risk of bias* |
|---|---|---|---|---|---|---|---|---|---|---|---|
| Koehler P et al, Germany, 2020 [63] | 19 | 62.6 | 60% | CAPA (5) | A. fumigatus (3) | VOR (2), ISA (1), CAS + VOR (2) | NM | Yes | NM | Dead (3) | Moderate |
| | | | | | | CAS (2): 70/50 mg/day, followed by IV VOR 6 & 4 mg/kg BID | | (all patients) | | Alive (2) | |
| | | | | | | IV VOR (2): (6/4 mg/kg) BID | | | | | |
| | | | | | | IV ISA (1): 200 mg TID followed by 200 mg OD | | | | | |
| Maes M et al, UK, 2021 [64] | 81 | 62 (50–70) | 59% | CAPA (3) | A. fumigatus (1) | Liposomal amphotericin (3) | NM | Yes | 15 | Death (1) | Moderate |
| | | | | | | | | (all patients) | (11–25) | Alive (2) | |
| Nasir N et al, Pakistan, 2020 [65] | 147 | 71 (51–85) | 77.7% | CAPA (9) | A. flavus/A. fumigatus (1) | AMB (2), VOR (3) | ≤28 days: (9) | Yes | 16 | Death (4) | Low |
| | | | | | A. fumigatus (1) | No antifungal (4) | | (all patients) | (6–35) | Alive (5) | |
| | | | | | A. flavus (4) | | | | | | |
| | | | | | A. flavus/A. niger (1), A. niger (2) | | | | | | |
| Bishburg E et al, USA, 2020 [66] | 89 | 63 (44–73) | 50% | CAC (8) | C.tropicalis (2), C.albicans (2), C.glabrata (2), C. parapsilosis (2) | CAS + FLU (4), FLU (3), CAS (1) | NM | Yes (all patients) | 40 (22–50) | Death (3) | Moderate |
| | | | | | | | | | | Alive (5) | |
| Lahmer T et al, Germany, 2021 [67] | 32 | 69.5 (27–84) | 72% | CAPA (11) | A. Fumigatus (9) | VOR (5), ISA (1), Liposomal AMB (5) | 19±3.5 | Yes | 18 | Death (4) | Moderate |
| | | | | | | | | (all patients) | (5–28) | Alive (7) | |
| Arastehfar A et al, Iran, 2021 [68] | 7 | 68 (27–75) | 42.8% | CAC (7) | C. albicans (4), | FLU + CAS (5) | NM | Yes | 33.5 | Death (6) | Low |
| | | | | | C. glabrata (3), | FLU (2) | | (all patients) | (7–83) | Alive (1) | |
| | | | | | R. mucilaginosa (1) | Loading dose: FLU 800 mg/day + CAS 70 mg/day (5), FLU 800 mg/day (2) | | | | | |
| | | | | | | Maintenance dose FLU 400 mg/day + CAS 50 mg/day (5), FLU 400 mg/day (2) | | | | | |

(*Continued*)

**Table 2.** (Continued)

| First author, Country, Year | Sample size | Age [mean ±sd/ Median (IQR)] years | Gender Male (%) | Diagnosis (Frequency) | Isolated fungal species in culture (Frequency) | Antifungal therapy (AFT) [Drug, dose, Frequency, RoA] | Duration of AFT (days) | #Managed with ICU care or MV support (Yes/No) | Duration of hospital stays [mean±sd/ Median (IQR)] days | Patient outcome (Frequency) | Overall Risk of bias* |
|---|---|---|---|---|---|---|---|---|---|---|---|
| Fekkar A et al, France, 2021 [69] | 7 | 55 (48–64) | 85.7% | CAPA (4), CAPA + CAM (2), CAPA + CAC (1) | A. fumigatus (5), F. proliferatum (1) | VOR 400 mg BID + | ≤28 days: (5) | Yes (all patients) | 30 (15–30) | Death (4) Alive (3) | Moderate |
| | | | | | | CAS 70 mg/day for 4 days (1) | | | | | |
| | | | | | | Liposomal amphotericine B 7 mg/kg/day for 6 days then Liposomal amphotericine B 7 mg/kg/day + CAS 70 mg/day for 18 days (1) | >28 days: (1) | | | | |
| | | | | | | VOR 400 mg BID for 9 days, then CAS 70 mg/day for 12 days (1) | | | | | |
| | | | | | | VOR 400 mg BID, then AMB 1 mg/kg/day, CAS 70 mg/day followed by ISA 200 mg/day (1) | | | | | |
| | | | | | | VOR 400 mg BID + CAS 70 mg/day, then AMB 1 mg/kg/day for 3 days (1) | | | | | |
| | | | | | | CAS 70 mg/day then VOR 300 mg (1) | | | | | |
| | | | | | | No AFT (1) | | | | | |
| Roman-Montes CM et al, Mexico, 2020 [70] | 14 | 48.5 (32–68) | 78.5% | CAPA (14) | A. fumigatus (6), A. flavus (1), A. niger (1), A. species (3) | VOR (10), ANI (2) No AFT (2) | ≤28 days: (6) >28 days: (5) NR (1) | Yes (all patients) | 30 | Death (8) Alive (5) Unknown (1) | Low |
| Segrelles-Calvo G et al, Spain, 2020 [71] | 7 | 58 (42–75) | 71.4% | CAPA (7) | A. fumigatus (3), A. flavus (2), A. niger (2) | IV ITR (2): 200 mg BID followed by 200 mg OD | ≤28 days: (3) >28 days: (2) | Yes (all patients) | 32.25 ± 14 | Death (5) Alive (2) | Low |
| | | | | | | IV ITR (1): 200 mg BID followed by 200 mg OD | | | | | |
| | | | | | | IV ITR (1): 200 mg OD | | | | | |
| | | | | | | IV AMB (1): 5 mg / kg / day | | | | | |
| | | | | | | No AFT (2) | | | | | |

(Continued)

**Table 2.** (Continued)

| First author, Country, Year | Sample size | Age [mean ±sd/ Median (IQR)] years | Gender Male (%) | Diagnosis (Frequency) | Isolated fungal species in culture (Frequency) | Antifungal therapy (AFT) [Drug, dose, Frequency, RoA] | Duration of AFT (days) | #Managed with ICU care or MV support (Yes/No) | Duration of hospital stays [mean±sd/ Median (IQR)] days | Patient outcome (Frequency) | Overall Risk of bias* |
|---|---|---|---|---|---|---|---|---|---|---|---|
| Mitaka H et al, USA, 2020 [72] | 4 | 78 (77–82) | 100% | CAPA (4) | A. species (3) | VOR (3), | ≤28 days: (3) | Yes (all patients) | 35 | Death (4) | Moderate |
| | | | | | A. fumigatus (1) | CAS (1) | >28 days: (1) | | | | |
| Salmanton-García J et al, Germany, 2021 [73] | 186 | 68 (58–73) | 72.6% | CAPA (158) | A. fumigatus (122), A. niger (13), A. flavus (10), A. terreus (6), A. calidoustus (1), A. lentulus (1), A. nidulans (1), A. penicillioides (1), A. versicolor (1), A. tubingensis (1), Aspergillus spp (1) | Liposomal AMB (23), Deoxycholate AMB (11), lipid complex AMB (2), ANI (10), CAS (13), MIC (1), IBR (1), VOR (98), ISA (23), POS (4), FLU (1) | NR | Yes (all patients) | NR | Death (119) Alive (39) | Low |
| Søgaard KK et al, Switzerland, 2021 [74] | 3 | 64.4 (50.4–74.2) | 61.1% | CAPA (2), CAC (1) | A. fumigatus (2) | FLU (1), CAS (3), ANI (1), VOR (2), | NM | Yes (all patients) | 7.7 (4.1–12.3) | NR | Moderate |
| | | | | | C. albicans (1) | | | | | | |
| Versyck M et al, France, 2021 [75] | 2 | 63.5 (55–72) | 100% | CAPA (2) | A. fumigatus (2) | VOR (2) | ≤28 days: (2) | Yes (all patients) | 15.2 (2–42) | Death (2) | Low |
| Salehi M et al, Iran, 2020 [76] | 53 | 63.1 (27–90) | 43.4% | Oropharyngeal CAC (53) | C. albicans (46), | FLU (21), NYS (13), CAS (1), FLU + NYS (17) | 4.79 ± 2.11 | NM clearly | NM | NR | Moderate |
| | | | | | C. glabrata (7), | | | | | | |
| | | | | | C dubliniensis (6), | No AFT (1) | | | | | |
| | | | | | C parapsilosis sensu stricto (3), | | | | | | |
| | | | | | C tropicalis (2), | | | | | | |
| | | | | | P kudriavzevii (1). | | | | | | |
| Buehler PK et al, Switzerland, 2020 [77] | 34 | 60 (54–69) | 77.8% | CAPA (5), CAC (29) | A. species (5), C. species (29) | AFT (10) | NM | Yes (all patients) | 24 | NR | Moderate |
| Seaton RA et al, UK, 2020 [78] | 13 | 71 (17–104) | 51.8% | Unknown fungal infection (13) | Not mentioned | CAS (7), FLU (5), VOR (1) | NR | NM clearly | NR | NR | Moderate |

#Patient who received ICU care or MV support during hospital stay, anytime

AFT: Antifungal therapy; AMB: Amphotericin B; ANI: Anidulafungin; BID: bis in die (twice daily); BSAA: Broad spectrum antifungal agents; CAS: Caspofungin; CAC: COVID-19 Associated Candidemia; CAPA: COVID-19 Associated Pulmonary Aspergillosis; CAM: COVID-19 Associated Mucormycosis; ECH: Echinocandins; FLU: Fluconazole; FLUC: Flucytosine; IBR: Ibrexafungin; ICU: Intensive care unit; ISA: Isavuconazole; ITR: Itraconazole; IV: Intravenous; MIC: Micafungin; MV: Mechanical ventilation; NM: Not mentioned; NR: Not reported; NYS: Nystatin; OD: Once in a day; POS: Posaconazole; TID: ter in die (Thrice daily); VOR: Voriconazole.

## 3.5. Prescription pattern of antifungal agents

**3.5.1. Case report and case series.** All patients in case reports (n = 38) and case series (n = 25) were prescribed with AFT. Azoles (26/38, 68.4%) were the most commonly used antifungal agents, followed by amphotericin B (18/38, 47.3%), echinocandins (12/38, 31.5%) and unknown antifungal agent (2/38, 5.2%) among patients in case reports. In patients with case series, Azoles (20/25, 80%) were the most commonly used antifungal agents, followed by echinocandins (9/25, 36%), and amphotericin B (3/25, 12%) (Table 1).

**3.5.2. Cohort studies.** Out of 416 patients in cohort studies, 393 were prescribed with AFT (either as monotherapy or as combination therapy). Azoles (251/393, 63.8%) were the most commonly used antifungal agents, followed by echinocandins (68/393, 17.3%), amphotericin B (66/393, 16.7%), unknown antifungal agent (53/393, 13.4%) and Ibrexafungerp (1/393, 0.25%) (Table 2).

## 3.6. Duration of Antifungal Therapy (AFT) and hospital stay

**3.6.1. Case report and case series.** Among 38 patients in case reports, a majority (n = 21, 55.2%) received AFT for ≤28 days, 11 patients (28.9%) for >28 days, and 6 patients (15.7%) the duration of AFT was not adequately reported. The total duration of hospital stays among the study patients ranged from 2 to 175 days. During hospital stay, the majority of patients were received ICU care or mechanical ventilation support. (Table 1).

**3.6.2. Cohort studies.** Out of 393 patients, 41 patients (10.4%) received the AFT for ≤28 days and 33 patients (8.3%) for >28 days. However, in 319 patients (81.1%) the duration of AFT was not adequately reported (Table 2).

## 3.7. All-cause mortality

In case reports 16 (42.1%) patients were died and 22 (57.9%) alive. In case series, 10 (40%) patients were died and 15 (60%) alive (Table 1). In cohort studies, 193 (46.3%) patients died, 103 (24.7%) were alive and 120 (28.8%) patients' status was unknown (Table 2).

Among 21 cohort studies, five studies [62, 65, 69–71] were included for meta-analysis and observed that the frequency of all-cause mortality was high among patients who did not receive any AFT [7/11, 63.6%] as compared to patient who received AFT [22/43, 51%] following fungal secondary infections. However, the pooled risk ratio showed that there was no significant difference in all-cause mortality between patients with AFT and without AFT [RR: 0.73, 95% CI: 0.46–1.15, p = 0.17, $I^2$ = 0%] [Fig 2(A)].

Three studies [62, 63, 68] were included in meta-analysis and found that the frequency of all-cause mortality was lower among patients who received combination AFT [7/13, 53.8%] as compared to the patients using monotherapy [9/14, 64.2%] for the management of fungal secondary infections. However, the pooled risk ratio showed that there was no significant difference in all-cause mortality between these groups [RR: 1.08, 95% CI: 0.48–2.43, p = 0.85, $I^2$ = 39%] [Fig 2(B)].

Four studies [69–72] were included in meta-analysis to assess the association between all-cause mortality and the duration (for ≤28 days vs >28 days) of AFT uses. The frequency of all-cause mortality was lower in patients who received AFT for >28 days [4/9, 44.4%] as compared to those who took AFT for ≤28 days [13/17, 76.4%]. However, there was no significant difference between the groups [RR: 0.87, 95% CI: 0.45–1.67, p = 0.67, $I^2$ = 15%] was observed as estimated by pooled risk ratio [Fig 2(C)].

Further, we included three studies [65, 70, 71] in meta-analysis to assess the all-cause mortality among CAPA patients with AFT and without AFT. Though the frequency of all-cause mortality was lower among CAPA patients who were on AFT [12/22, 54.5%] as compared to

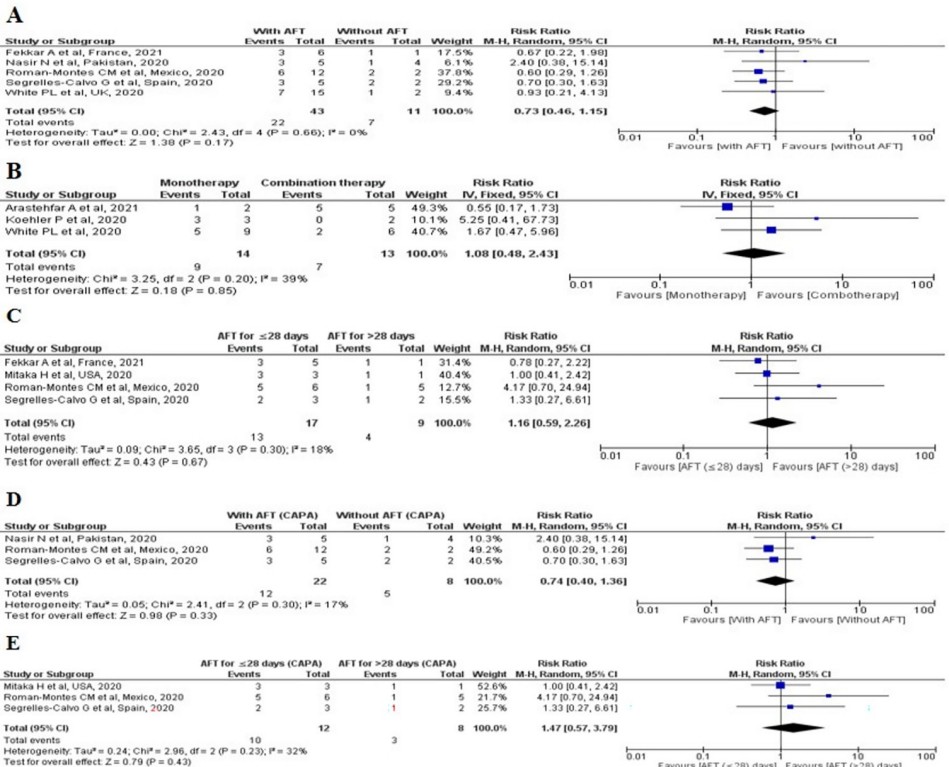

**Fig 2. All-cause mortality associated with fungal secondary infections among COVID-19 patients who used AFT.**
(A): All-cause mortality associated with AFT and without AFT in fungal secondary infections among COVID-19 patients; (B): All-cause mortality associated with mono- and combination AFT in fungal secondary infections among COVID-19 patients; (C): All-cause mortality associated with duration of AFT in fungal secondary infections among COVID-19 patients; (D): All-cause mortality associated with AFT and without AFT in patients with CAPA and (E): All-cause mortality associated with duration of AFT in patients with CAPA.

those who weren't on AFT [5/8, 62.5%], the pooled risk ratio revealed no significant difference between the groups [RR: 0.74, 95% CI: 0.40–1.36, p = 0.33, $I^2$ = 17%] [Fig 2(D)]. The meta-analysis of three studies [70–72] that assessed the association between the all-cause mortality and duration of AFT (for ≤28 days vs >28 days) among CAPA patients showed that the frequency of all-cause mortality was lower in patients who were on AFT for >28 days as compared to those used AFT for ≤28 days. However, the pooled risk-ratio revealed no significant difference between the groups [RR: 1.47, 95% CI: 0.57–3.79, p = 0.43, $I^2$ = 32%] [Fig 2(E)].

The summary of fungal secondary infections, AFT used, duration of AFT, and outcomes among COVID-19 patients is presented in S2 Table.

## 4. Discussion

The worldwide mortality associated with COVID-19 is 3.97 million [13]. However, there is no published literature suggesting the global mortality in COVID-19 patients with fungal secondary infections. In this review, the prevalence of fungal secondary infections (cohort studies) was 28.2% [416/1474] and the overall all-cause mortality rate in patients with COVID-19 associated fungal secondary infections was 45.7% [219/479]. Further, the all-cause mortality associated with CAPA and CAM was 75.2% [198/308] and 13% [2/15] respectively. The mortality rate associated with CAPA was lower (51.2%) in a recent review published by Singh S et al., [14] another recent review reported the mortality associated with CAM as 30.7% [15]. The

mortality associated with CAC was not adequately reported in the studies that we reviewed, however a recently published study from Atlanta reported that CAC associated mortality was up to 30.9% [16]. Thus, the overall mortality rates associated with fungal secondary infections in COVID-19 patients are higher as compared to COVID-19 alone. There could be various contributing factors for this such as type of fungal species, multiple fungal secondary infections, AFT used, presence of other bacterial or viral superinfections, use of immunosuppressive therapy, presence of other co-morbid conditions, and age of the patients.

CAPA (n = 36 studies) [23–35, 37–42, 55–58, 61, 63–65, 67, 69–75, 77] was the most commonly diagnosed fungal co-infection, followed by CAC (n = 9 studies) [49, 58, 60, 66, 68, 69, 74, 76, 77] and CAM (n = 9 studies) [18, 43–48, 59, 69]. Many studies from Europe, Australia and China have reported increased prevalence of CAPA (range: 20–35%) [17]. We observed that most of the studies including patients with CAPA (n = 22 studies) and CAC (n = 5 studies) were reported from Europe. *Aspergillus Fumigatus* was the most common causative organism identified through culture media in these studies. Voriconazole is the recommended first-line antifungal therapy whereas, amphotericin B is the second-line agent for CAPA [17]. We observed that the most common AFT prescribed for CAPA was voriconazole followed by amphotericin B. The recommended maintenance doses of voriconazole and amphotericin B are 200 mg bid [18] and 3mg/kg/day respectively [19]. In our review, the prescribed dose range of voriconazole was 200 to 800 mg/day and amphotericin B was 3 to 10 mg/kg/day (either as single-drug therapy or as combination). The recommended median duration of AFT is 76 days [18], our review explored that the total duration of AFT ranged from 2 to 90 days.

The prevalence of CAC was high among the Chinese population (23.5%) [20]. The recommended AFT includes echinocandins, azoles and amphotericin B [21]. In this review, we observed that the common causative organism of CAC was *Candida albicans* and the most common AFT prescribed for CAC was fluconazole followed by nystatin. Fluconazole was prescribed at a dose of 400 mg/day, and the total duration of AFT ranged from 4 to 21 days.

We observed most of the published literature for CAM were from India (n = 5 studies) [18, 43, 46, 48, 59]. The most common AFT used for the management of CAM was amphotericin B either as a single drug or in combination with other antifungal drugs. The current guideline for the management of mucormycosis recommends liposomal amphotericin B and posaconazole as the first-line AFT [2]. The recommended dose of amphotericin B is 5 to 10 mg/kg/day [22], we observed that in the reviewed studies amphotericin B was used in the dose range of 3 to 5 mg/kg/day. The total duration of AFT for CAM was ranged from 11 to 150 days.

There were 33 studies that included only CAPA patients [23–42, 55–57, 61, 63–65, 67, 70–73, 75]. Among them 17 studies used single-drug therapy, eleven studies used combination AFT, three studies used both single-drug and combination AFT, and two studies used AFT the details of which were not adequately reported.

Among nine studies [18, 43–48, 59, 60] that reported only CAM, two studies used liposomal amphotericin B alone, five studies used combination AFT (liposomal amphotericin B with azoles, liposomal amphotericin B with caspofungin and azole, voriconazole and caspofungin) and one study used both single-drug as well as combination AFT. It was observed that all the patients in the studies, where single-drug therapy was used were alive. Among seventeen patients that received combination therapy, five were dead and remaining were alive.

There were five studies [49, 54, 60, 66, 68] that included only CAC patients. Two studies used single-drug therapy, two studies used both single-drug as well as combination AFT and one study used AFT the details of which are not adequately reported. In two studies where single-drug therapy was used all the patients were alive, whereas in other studies the mortality details were not adequately reported.

The results of meta-analysis revealed that there was no significant difference in terms of all-cause mortality among patients who received AFT & did not receive AFT ($p = 0.17$), all-cause mortality & type of AFT used ($p = 0.85$), and all-cause mortality & duration of AFT ($p = 0.87$). There could be various confounding factors such as delay in diagnosis of fungal secondary infections in earlier or terminal stages of COVID-19 by physicians, diagnostic difficulties in mycological detection, increased risk of bacterial or viral infections in short to long term of infections, presence of polymorbidity and low sample size might be the reasons for the non-significance differences found in all-cause mortality with who received & did not receive AFT, type of AFT used and duration of AFT.

However, the survival frequency was high among patients using AFT [21/43, 48.8%] as compared to those who didn't use AFT [4/11. 36.4%], the patients using combination AFT [6/13, 46.2%] as compared to those who were using a single antifungal drug [5/14, 35.8%] and among patients using AFT for >28 days [5/9, 55.5%] as compared to those who were using AFT for ≤28 days [4/17, 23.5%].

Further, the sub-group analysis including studies that reported CAPA patients alone, revealed that was no significant difference in terms of all-cause mortality among patients who received AFT & did not receive AFT ($p = 0.33$), and all-cause mortality & duration of AFT ($p = 0.43$). However, in CAPA patients also, we observed a high survival frequency among patients who used AFT [10/22, 45.4%] and when AFT was used for >28 days [5/8, 62.5%]. However, we couldn't find the studies for similar subgroup meta-analysis in patients with CAC and CAM.

At the time of literature search, there was no published literature available on randomized control studies conducted among patients with fungal secondary infections associated with COVID-19. However, we made an attempt to explore if there are any such ongoing studies. Our search revealed that currently there are only two ongoing studies. One of which is phase 2 and another one is phase 3 study. The expected date of completion of these studies will be first quarter of 2022. The availability of these study results will hopefully add to the existing evidence of efficacy of AFT in treating fungal secondary infections among COVID-19 patients. The details of these ongoing studies are presented in *S5 Table*.

## 4.1. Limitations

We could not able to establish the efficacy of any individual AFT or class of antifungal agent/s that are used for the treatment of fungal secondary infections in COVID-19 patients due to a lack of adequate data reporting among the included studies about antifungal regimen. It was observed that many studies were reported antifungal drugs without complete information about antifungal regimens including doses, frequency and duration. In addition, at the time of literature search, there was no published literature on randomized control studies conducted in COVID-19 patients with fungal secondary infections. Availability of such literature would have added more clarity on efficacy of AFT in different fungal secondary infections associated with COVID-19 patients.

## 5. Conclusion

The prevalence of fungal secondary infections among COVID-19 patients was 28.2%. The most common fungal secondary infections among COVID-19 patients were CAPA, CAC and CAM. Voriconazole, fluconazole and liposomal amphotericin B were the most commonly used antifungal agents for the management of CAPA, CAC and CAM respectively. The results of this systematic review and meta-analysis suggest that the survival frequency was high among patients who were; on AFT for the management of fungal secondary infections, using

combination AFT and using AFT for >28 days. However, the pooled risk ratio, revealed that the all-cause mortality in COVID-19 patients with fungal secondary infections is not associated with the type and duration of AFT may be due to the availability of confounding factors such as delay in diagnosis of fungal secondary infections, presented with multiple comorbidities, older age and a small number of events that may reduced power to detect a difference, may contribute for outcomes in such patients.

## Supporting information

**S1 Checklist. PRISMA checklist for systematic reviews (2009).**
(DOCX)

**S1 Appendix. Details of PICOS format for study inclusion criteria.**
(DOCX)

**S2 Appendix. Details on search strategies applied in various.**
(DOCX)

**S1 Table. Details of excluded literatures form the review.**
(DOCX)

**S2 Table. Summary of fungal secondary infections, antifungal therapy (AFT) used, duration of AFT and outcomes COVID-19 patients.**
(DOCX)

**S3 Table. Risk of bias assessment for case report and case series using methodological quality and synthesis.**
(DOCX)

**S4 Table. Risk of bias assessment for cohort studies using newcastle-ottawa quality assessment.**
(DOCX)

**S5 Table. Details of ongoing randomized control studies involving COVID-19 patients with fungal secondary infections.**
(DOCX)

## Author Contributions

**Conceptualization:** Sujit Kumar Sah, Atiqulla Shariff, Niharika Pathakamuri.

**Data curation:** Sujit Kumar Sah, Atiqulla Shariff, Niharika Pathakamuri, Subramanian Ramaswamy, Madhan Ramesh, Krishna Undela, Malavalli Siddalingegowda Srikanth, Teggina Math Pramod Kumar.

**Formal analysis:** Sujit Kumar Sah, Atiqulla Shariff, Niharika Pathakamuri, Subramanian Ramaswamy, Madhan Ramesh, Krishna Undela, Malavalli Siddalingegowda Srikanth, Teggina Math Pramod Kumar.

**Funding acquisition:** Sujit Kumar Sah, Atiqulla Shariff.

**Investigation:** Sujit Kumar Sah, Atiqulla Shariff, Niharika Pathakamuri, Subramanian Ramaswamy, Krishna Undela, Malavalli Siddalingegowda Srikanth.

**Methodology:** Sujit Kumar Sah, Atiqulla Shariff.

**Project administration:** Sujit Kumar Sah, Atiqulla Shariff, Subramanian Ramaswamy, Madhan Ramesh, Krishna Undela, Malavalli Siddalingegowda Srikanth, Teggina Math Pramod Kumar.

**Resources:** Sujit Kumar Sah, Atiqulla Shariff, Niharika Pathakamuri, Subramanian Ramaswamy, Madhan Ramesh, Krishna Undela, Teggina Math Pramod Kumar.

**Software:** Sujit Kumar Sah, Atiqulla Shariff.

**Supervision:** Subramanian Ramaswamy, Madhan Ramesh, Krishna Undela, Malavalli Siddalingegowda Srikanth, Teggina Math Pramod Kumar.

**Validation:** Sujit Kumar Sah, Atiqulla Shariff, Niharika Pathakamuri.

**Visualization:** Sujit Kumar Sah, Atiqulla Shariff, Niharika Pathakamuri, Subramanian Ramaswamy, Madhan Ramesh, Krishna Undela, Malavalli Siddalingegowda Srikanth, Teggina Math Pramod Kumar.

**Writing – original draft:** Sujit Kumar Sah, Atiqulla Shariff, Niharika Pathakamuri, Madhan Ramesh.

**Writing – review & editing:** Sujit Kumar Sah, Atiqulla Shariff, Niharika Pathakamuri, Subramanian Ramaswamy, Madhan Ramesh, Krishna Undela, Malavalli Siddalingegowda Srikanth, Teggina Math Pramod Kumar.

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
