## [Decision Letter · Decision Letter 0]

31 Jan 2022

PONE-D-21-30319Antifungal therapy in the management of fungal co-infections in COVID-19 patients: A systematic review and meta-analysisPLOS ONE

Dear Dr. Ramaswamy,

Thank you for submitting your manuscript to PLOS ONE. After careful consideration, we feel that it has merit but does not fully meet PLOS ONE’s publication criteria as it currently stands. Therefore, we invite you to submit a revised version of the manuscript that addresses the points raised during the review process.

The reviewers have raised some valid points that should be addressed. Some points may be satisfactorly explained in the response but others require adjustments in the manuscript. The manuscript could also use a final proofreading.

We look forward to receiving your revised manuscript.

Kind regards,

Joy Sturtevant

Academic Editor

PLOS ONE

Journal Requirements:

Whilst you may use any professional scientific editing service of your choice, PLOS has partnered with both American Journal Experts (AJE) and Editage to provide discounted services to PLOS authors. Both organizations have experience helping authors meet PLOS guidelines and can provide language editing, translation, manuscript formatting, and figure formatting to ensure your manuscript meets our submission guidelines. To take advantage of our partnership with AJE, visit the AJE website (http://aje.com/go/plos) for a 15% discount off AJE services. To take advantage of our partnership with Editage, visit the Editage website (www.editage.com) and enter referral code PLOSEDIT for a 15% discount off Editage services.  If the PLOS editorial team finds any language issues in text that either AJE or Editage has edited, the service provider will re-edit the text for free.

The funders had no role in study design, data collection and analysis, decision to publish, or preparation of the manuscript

Reviewers' comments:

Reviewer's Responses to Questions

**Comments to the Author**

1. Is the manuscript technically sound, and do the data support the conclusions?

Reviewer #1: No

Reviewer #2: Yes

2. Has the statistical analysis been performed appropriately and rigorously? 

Reviewer #1: Yes

Reviewer #2: Yes

3. Have the authors made all data underlying the findings in their manuscript fully available?

Reviewer #1: Yes

Reviewer #2: Yes

4. Is the manuscript presented in an intelligible fashion and written in standard English?

Reviewer #1: No

Reviewer #2: Yes

5. Review Comments to the Author

Reviewer #1: The authors have performed a systematic review and meta-analysis of studies evaluating the prevalence and outcomes of fungal co-infections in patients with COVID-19. While this is an important potential sequelae of COVID-19 to investigate further, the analysis seems to focus heavily on individual patient data from an extremely small number of studies and patients. This makes it nearly impossible to draw any conclusions from these analyses. For an IPD meta-analysis, additional data may be required by requesting it from authors and following the PRISMA-IPD recommendations. In the absence of more robust individual patient data, the study could focus exclusively on prevalence of fungal co-infections without drawing conclusions about comparative outcomes in these patients.

Some suggestions for consideration:

1. The funding statement is not clear, was funding received for this manuscript and if so, from which entity?

2. It should be made clear in the abstract and earlier in the methods that these study includes an individual patient meta-analysis. Perhaps it is also necessary to include criteria for inclusion into the SR vs. into the IPD meta-analysis, as these appear to be slightly different populations in this study. For example, did studies have to have IPD in order to be included in the SR? Perhaps PRISMA-IPD checklist should be used for this study?

3. Please provide a justification for why only English language studies were searched given the global impact of COVID-19 and potential sequelae.

4. Minor: please correct spelling of clinicaltrial.gov

5. Please clarify whether full text screening was performed in duplicate.

6. How was missing data handled? Presumably some studies will report prevalence of anti-fungal use but not on patient outcomes.

7. Are the authors able to stratify patients by those with fungal co-infections on admission vs. later in hospital stay (e.g. > 48h)? This could be helpful to delineate outcomes between "co-infections" and "secondary or nosocomial infections".

8. The search yield seems extremely low given the volume of studies on COVID-19 up to present. Can the authors explain why they applied "Review, Systematic review" filters in pubmed if they were not aiming to include these studies. The pubmed filters seem to narrow the results significantly.

9. Please indicate a measure of patient severity, e.g., how many in ICU or on mechanical ventilation?

10. Were any studies (cohort) included that exclusively looked at patients with fungal infections? The rate of fungal infections seems high and could be elevated based on the denominator selected.

11. Can the authors provide additional detail on the microbiological methods used for detection of fungi, as well as the clinical criteria used to identify true infections?

12. How many patients were eligible for IPD meta-analysis? The numbers in forest plots seem very low and not conducive to drawing conclusions on outcomes in this patient population.

13. Comparing all-cause mortality among type and duration of AFT doesn't seem appropriate given 1) the low sample size and 2) the lack of accounting for confounding factors.

Reviewer #2: The present systematic review and meta-analysis aims to evaluate the prevalence of fungal co-infections among COVID-19 patients and their recovery or all-cause mortality following antifungal therapy. After analyzing the selected reports, the paper indicates that there is a high survival among patients who used antifungal drugs, and that the all-cause mortality in COVID-19 patients with fungal coinfections is not associated with type and duration of the drug therapy. The work is interesting and relevant; however, the manuscript has some points, that need to be adjusted.

A critical point is the use of “unpublished material”. This kind of data is not easily available to anyone. So, what is the rational for using material that has not been peer reviewed and accessible to any researcher? Unless is from bioRxiv, for example. This needs to be very well clarified. Also, the Results section is extremely long and should be cut.

More points to be addressed:

Introduction

- “The exact incidence of fungal co-infections in COVID-19 patients has not been established.” – this reviewer does not agree with this sentence. Please check these works, where this has been estimated (in general or locally): doi: 10.1007/s11046-020-00462-9; DOI: 10.3390/jof7090720; doi: 10.1016/j.cmi.2020.06.025 and correct the information;

- In terms of drugs, I would recommend checking this recent work: doi: 10.1016/j.jmii.2020.05.013;

- “The commonly used antifungal therapy (AFT) include liposomal amphotericin B, azole antifungals,and echinocandins[6].” – antifungals after “azoles” is unnecessary. All examples are antifungals. Correct this in the entire MS;

M&M:

- RevMan 5.4 – country and manufacturer?

Results:

- This section is too long. The MS provides nice tables, so some of the information should be summarized. Is too exhaustive.

6. PLOS authors have the option to publish the peer review history of their article (what does this mean?). If published, this will include your full peer review and any attached files.

Reviewer #1: No

Reviewer #2: No

---

## [Author Response · Author response to Decision Letter 0]

26 Apr 2022

Reviewer 1

1. The funding statement is not clear, was funding received for this manuscript and if so, from which entity?

Response: Attended- The funding details have been updated in the revised manuscript and highlighted. [Page no. 18; line number: 413-414]

2. It should be made clear in the abstract and earlier in the methods that these study includes an individual patient meta-analysis. Perhaps it is also necessary to include criteria for inclusion into the SR vs. into the IPD meta-analysis, as these appear to be slightly different populations in this study. For example, did studies have to have IPD in order to be included in the SR? Perhaps PRISMA-IPD checklist should be used for this study?

Answer: Attended: We appreciate your suggestion for looking to IPD meta-analysis, which was new to us. We went through various related articles and found that IPD meta-analysis is also used to retrieve unpublished information from clinical studies (especially clinical trials comparing interventions). However, in our study, different populations and study designs were used but we could not able to find any clinical trial-related studies during data extraction. We included all published data except one study (reference number 77). For that one study, we contacted the original study author and gathered information. Soon after that with the same information that article got published. 

In this review, we tried to compare the outcomes within the cohort studies which may create scope for the IPD meta-analysis. With concern that as well as reviewer suggestions we contacted selected articles’ authors through the mail but their response was very poor. To date, they were not provided information as the information was published in cohort studies.

Due to these difficulties in this review, we were unable to include IPD meta-analysis but added a few critical points in the conclusion that may be helpful to resolve this problem. 

3. Please provide a justification for why only English language studies were searched given the global impact of COVID-19 and potential squeal.

Response: Attended- The main reason behind selecting only the English language was that English is most commonly readable, understandable and easily reachable to international researchers. Further, selecting multiple language articles may require personnel who were multiple language experts, which is difficult to get. Therefore, we selected the standard language English, which is spoken and written by the majority of researchers around the globe. 

4. Minor: please correct spelling of clinicaltrial.gov

Response: Attended-Spelling of ClinicalTrial.gov has been corrected in the method section of the abstract [page no. 3; line number: 69] and in the main manuscript [page no. 6; line number: 146], same has been highlighted in the revised manuscript. 

5. Please clarify whether full text screening was performed in duplicate.

Response: Attended-The title of the identified records was screened during duplication removal. However, both title and full-text were reviewed for the study potential records before removing duplicates.

6. How was missing data handled? Presumably some studies will report prevalence of anti-fungal use but not on patient outcomes.

Response: Attended-If any study relevant data was missing or incomplete then review authors were engaged to search original sources or contacted original authors through mail to obtain any missing information. If it is not possible to obtain missing information, then those studies were excluded from this review.

7.Are the authors able to stratify patients by those with fungal co-infections on admission vs. later in hospital stay (e.g. > 48h)? This could be helpful to delineate outcomes between "co-infections" and "secondary or nosocomial infections".

Response: Attended-Initially we attend to stratify patients by those with fungal co-infections on admission vs later in hospital stay but we found major issues with many patients that they had more than one hospital admission and delay in detection of antifungal co-infections. Most of the time patients developed multiple coinfections together and made difficulties in obtaining the exact time of fungal infections. These made it difficult in obtaining exact information.

8.The search yield seems extremely low given the volume of studies on COVID-19 up to present. Can the authors explain why they applied "Review, Systematic review" filters in pubmed if they were not aiming to include these studies. The pubmed filters seem to narrow the results significantly.

Response: Attended-During PubMed's initial search we included Review and Systematic review terms search to extract the maximum volume of studies. We included all the fields during the search in PubMed and inadvertently mentioned them in a confusing way. Clarity has been provided and updated in the revised manuscript. 

Corrections have been made and uploaded in Supplementary Appendix S2 and highlighted. [Supplementary file: Page number 5]

9. Please indicate a measure of patient severity, e.g., how many in ICU or on mechanical ventilation?

Response: Attended- Most of the patients from included studies were admitted to ICU or received mechanical ventilation during their hospital stay. As per the reviewer's suggestion, we authors attempted to address this and updated in table 1 and table 2. 

The details are updated in the result section [page no. 11; line number: 262-263] and in table 1 and table 2. The same has been highlighted in the revised manuscript. 

10. Were any studies (cohort) included that exclusively looked at patients with fungal infections? The rate of fungal infections seems high and could be elevated based on the denominator selected.

Response: Attended- The selected cohort study were included patients with fungal infections and evaluated outcomes in various domains such as identifying prevalence, the effectiveness of the diagnostic test, duration of hospital stay, ICU admission, clinical features, diagnostic tests, mortality, co-infections and their management. However, exactly no studies were evaluated the efficacy of antifungal agents in fungal co-infection patients with COVID-19. We rigorously analysed the data from each study and draw the correct information.

11. Can the authors provide additional detail on the microbiological methods used for detection of fungi, as well as the clinical criteria used to identify true infections?

Response: Attended- the details have been updated in the methods section and highlighted in the revised manuscript. [page no. 6; line number: 136-137] 

12. How many patients were eligible for IPD meta-analysis? The numbers in forest plots seem very low and not conducive to drawing conclusions on outcomes in this patient population.

Response: Attended- Please look for the responses in query 2. However, we added a few critical points in conclusion to resolve this issue. 

The details have been uploaded in the conclusion section of the abstract [page no. 4; line number: 83, 85 and 87-89] and in the main manuscript. [page no. 17; line number: 409-411]

13. Comparing all-cause mortality among type and duration of AFT doesn't seem appropriate given 1) the low sample size and 2) the lack of accounting for confounding factors.

Response: Attended-The details have been updated and highlighted in the revised manuscript. [page no. 15-16; line number: 362-367]

Reviewer-2

1. A critical point is the use of “unpublished material”. This kind of data is not easily available to anyone. So, what is the rational for using material that has not been peer reviewed and accessible to any researcher? Unless is from bioRxiv, for example. This needs to be very well clarified.

Response: Attended-We authors were aware that data from unpublished studies can itself introduce bias to the study. However, the current study was emerging and multiple publications were published on pre-print websites before getting peer-reviewed. This publication may have study-related important information, which can be obtained by contacting the original authors. 

In the current study, we tried to gather data from all types of possible databases to strengthen the search strategies. However, the record which we identified from unpublished databases were excluded during screening itself due to inclusion criteria except one (reference number 77). For that one study, we contacted the original study author and gathered information. Soon after that with the same information that article got published. 

.https://doi.org/10.1016/j.xcrm.2021.100229

The details about how we gathered missing information have been updated and highlighted in the revised manuscript. [page no. 7; line number: 158-160]

2. Introduction

- “The exact incidence of fungal co-infections in COVID-19 patients has not been established.” – this reviewer does not agree with this sentence. Please check these works, where this has been estimated (in general or locally): doi: 10.1007/s11046-020-00462-9; DOI: 10.3390/jof7090720; doi: 10.1016/j.cmi.2020.06.025 and correct the information;

- In terms of drugs, I would recommend checking this recent work: doi: 10.1016/j.jmii.2020.05.013;

Response: Attended-The sentence has been re-structured after correction. The details have been highlighted in the revised manuscript. [page no. 5; line number: 109-111]

Reference: Information was taken from reference number 2. 

Song G, Liang G, Liu W. Fungal Co-infections Associated with Global COVID-19 Pandemic: A Clinical and Diagnostic Perspective from China. Mycopathologia. 2020;185(4): 599–606. doi:10.1007/s11046-020-00462-9

3. - “The commonly used antifungal therapy (AFT) include liposomal amphotericin B, azole antifungals,and echinocandins[6].” – antifungals after “azoles” is unnecessary. All examples are antifungals. Correct this in the entire MS.

Responses: Attended- This issue was corrected throughout the manuscript. AAs per the reviewer's suggestion, the word antifungal after azole has been removed throughout the manuscript in the revised manuscript. 

a. [Page number 5, line number 123]

b. [Page number 10, line number 246]

c. [Page number 11, line number 249]

d. [Page number 11, line number 252]

e. [Page number 15, line number 350]

4. M&M: RevMan 5.4 – country and manufacturer?

Response: Attended-The details have been updated and highlighted in the revised manuscript in the statistical analysis section of methods. [page no. 8; line number: 184-186]

5. Results:

- This section is too long. The MS provides nice tables, so some of the information should be summarized. Is too exhaustive.

Response: Attended-We presented results in various possible ways. However, due to the reviewer's suggestion, few sentences were re-structured to reduce the length of the result section without altering the main findings. 

The sentences have been re-structured and highlighted in the revised manuscript. 

[page no. 11; line number: 260]

[page no. 12; line number: 271-172]

[page no. 12; line number: 277-278]

---Grammar, spelling and punctuation has been corrected throughout the manuscript and highlighted in the revised manuscript

Response to editor

1. We notice that your manuscript file was uploaded on Sep 19 2021. Please can you upload the latest version of your revised manuscript as the main article file.

Attended: Changes has been made and uploaded revised manuscript without tracker changes. 

2.Please include your amended statements within your cover letter.

Attended: Included in cover letter and uploaded. 

3. PLOS requires an ORCID iD for the corresponding author. 

Attended: Added ORCID ID of corresponding author.

4. Your article cannot proceed until you upload a copy of the completed PRISMA checklist as Supporting Information. We note that this manuscript is a systematic review or meta-analysis; our author guidelines therefore require that you use PRISMA guidance to help improve reporting quality of this type of study.

Attended: Prisma check list has been uploaded along with the revised supporting file. [Page number: 2-3].

---

## [Decision Letter · Decision Letter 1]

10 May 2022

PONE-D-21-30319R1Antifungal therapy in the management of fungal co-infections in COVID-19 patients: A systematic review and meta-analysisPLOS ONE

Dear Dr. Ramaswamy,

Thank you for submitting your manuscript to PLOS ONE. After careful consideration, we feel that it has merit but does not fully meet PLOS ONE’s publication criteria as it currently stands. Therefore, we invite you to submit a revised version of the manuscript that addresses the points raised during the review process.

The reviewers and myself are pleased with the response to the previous critiques. There are a few minor points that still need to be addressed and are stated by Reviewer #1.

We look forward to receiving your revised manuscript.

Kind regards,

Joy Sturtevant

Academic Editor

PLOS ONE

Journal Requirements:

Reviewers' comments:

Reviewer's Responses to Questions

**Comments to the Author**

1. If the authors have adequately addressed your comments raised in a previous round of review and you feel that this manuscript is now acceptable for publication, you may indicate that here to bypass the “Comments to the Author” section, enter your conflict of interest statement in the “Confidential to Editor” section, and submit your "Accept" recommendation.

Reviewer #1: (No Response)

Reviewer #2: All comments have been addressed

2. Is the manuscript technically sound, and do the data support the conclusions?

Reviewer #1: Yes

Reviewer #2: Yes

3. Has the statistical analysis been performed appropriately and rigorously? 

Reviewer #1: Yes

Reviewer #2: Yes

4. Have the authors made all data underlying the findings in their manuscript fully available?

Reviewer #1: No

Reviewer #2: Yes

5. Is the manuscript presented in an intelligible fashion and written in standard English?

Reviewer #1: No

Reviewer #2: Yes

6. Review Comments to the Author

Reviewer #1: Thank you for making these improvements. Some additional, mostly minor suggestions:

Consider the term "fungal co-infection or secondary infection" rather than "co-infection" since they cannot be differentiated/disaggregated in this study. Co-infection generally refers to upon presentation or <48h after presentation with COVID-19, and it would be unlikely that all patients with fungal co-infection developed early after presentation.

Please clarify if any, and how many, studies included patients with suspected but not confirmed COVID-19. The methods indicate that patients must have had confirmed COVID-19, but for example Seaton et al, included many patients without positive SARS-CoV-2 tests. 10.1016/j.jinf.2020.09.024

The Seaton study is referenced by the authors as occurring in UK and Germany, but the study itself only indicates UK.

Thank you for adding a statement on limitations line 407-409, but this should be in limitations rather than conclusions. Line 408-409, please rephrase as a small number of events is not a confounding factor, but rather leads to reduced power to detect a difference when in fact one may exist.

Reviewer #2: (No Response)

7. PLOS authors have the option to publish the peer review history of their article (what does this mean?). If published, this will include your full peer review and any attached files.

Reviewer #1: No

Reviewer #2: No

---

## [Author Response · Author response to Decision Letter 1]

5 Jul 2022

Response to Reviewer:

Q.1. Consider the term "fungal co-infection or secondary infection" rather than "co-infection" since they cannot be differentiated/disaggregated in this study. Co-infection generally refers to upon presentation or <48h after presentation with COVID-19, and it would be unlikely that all patients with fungal co-infection developed early after presentation.

Response: Attended- We appreciate your suggestion and found using “fungal secondary infections” is suitable medical term instead of “fungal co-infections” in this study and the same has been changed throughout the manuscript. 

The changes have been updated in the revised manuscript and highlighted in the title section, abstract, Introduction, Methods, Results, Discussion, Conclusion, Figure and supplementary file.

Q2. Please clarify if any, and how many, studies included patients with suspected but not confirmed COVID-19. The methods indicate that patients must have had confirmed COVID-19, but for example Seaton et al, included many patients without positive SARS-CoV-2 tests. 10.1016/j.jinf.2020.09.024

Response: Attended- In this study, we only included the 479 patients out of 1537 COVID 19 patients, who was confirmed case of COVID-19 and diagnosed with fungal secondary infections. Hence, we excluded all the patients who were not confirmed cases of COVID-19. Majority of selected Cohort studies included patients with suspected but not confirmed COVID-19.

The details have been previously addressed in the result section “3.4. Participants’ characteristics and clinical diagnosis”. [Page 10, line number: 231-233]

Q3. The Seaton study is referenced by the authors as occurring in UK and Germany, but the study itself only indicates UK.

Response: Attended- Even though the authors are from various countries, this study was only conducted in the UK. Inadvertently added in result section in Study characteristics. 

The issue has been corrected and highlighted in the main manuscript in the revised manuscript. [Page 9, line number: 213]

Q4. Thank you for adding a statement on limitations line 407-409, but this should be in limitations rather than conclusions. Line 408-409, please rephrase as a small number of events is not a confounding factor, but rather leads to reduced power to detect a difference when in fact one may exist.

Response: Attended- The sentence has been rephrased and highlighted in the revised manuscript. 

[Page 17, line number: 412-413]

Editorial Response:

Q1. Funding details in cover letter?

Response: Attended- Funding details had been added in Cover letter and same uploaded during revision.

---

## [Editor Report · Decision Letter 2]

8 Jul 2022

Antifungal therapy in the management of fungal secondary infections in COVID-19 patients: A systematic review and meta-analysis

PONE-D-21-30319R2

Dear Dr. Ramaswamy,

We’re pleased to inform you that your manuscript has been judged scientifically suitable for publication and will be formally accepted for publication once it meets all outstanding technical requirements.

Kind regards,

Joy Sturtevant

Academic Editor

PLOS ONE
---

## [Editor Report · Acceptance letter]

18 Jul 2022

PONE-D-21-30319R2 

Antifungal therapy in the management of fungal secondary infections in COVID-19 patients: A systematic review and meta-analysis 

Dear Dr. Ramaswamy:

I'm pleased to inform you that your manuscript has been deemed suitable for publication in PLOS ONE. Congratulations! Your manuscript is now with our production department. 

Kind regards, 

on behalf of

Dr. Joy Sturtevant 

Academic Editor

PLOS ONE